OBSERVATION

# Host Polyunsaturated Fatty Acids Potentiate Aminoglycoside Killing of *Staphylococcus aureus*

William N. Beavers,[a,b] Matthew J. Munneke,[a] Alex R. Stackhouse,[b] Jeffrey A. Freiberg,[a,c] Eric P. Skaar[a,d,e]

aDepartment of Pathology, Microbiology, and Immunology, Vanderbilt University Medical Center, Nashville, Tennessee, USA

bDepartment of Pathobiological Sciences, Louisiana State University and Agricultural and Mechanical College, School of Veterinary Medicine, Baton Rouge, Louisiana, USA

cDivision of Infectious Diseases, Department of Medicine, Vanderbilt University Medical Center, Nashville, Tennessee, USA

dVanderbilt Institute for Infection, Immunology, and Inflammation, Vanderbilt University Medical Center, Nashville, Tennessee, USA

eVanderbilt Institute for Chemical Biology, Vanderbilt University, Nashville, Tennessee, USA

**ABSTRACT** Aminoglycoside antibiotics rely on the proton motive force to enter the bacterial cell, and facultative anaerobes like *Staphylococcus aureus* can shift energy generation from respiration to fermentation, becoming tolerant of aminoglycosides. Following this metabolic shift, high concentrations of aminoglycosides are required to eradicate *S. aureus* infections, which endangers the host due to the toxicity of aminoglycosides. Membrane-disrupting molecules prevent aminoglycoside tolerance in *S. aureus* by facilitating passive entry of the drug through the membrane. Polyunsaturated fatty acids (PUFAs) increase membrane permeability when incorporated into *S. aureus*. Here, we report that the abundant host-derived PUFA arachidonic acid increases the susceptibility of *S. aureus* to aminoglycosides, decreasing the aminoglycoside concentration needed to kill *S. aureus*. We demonstrate that PUFAs and aminoglycosides synergize to kill multiple strains of *S. aureus*, including both methicillin-resistant and -susceptible *S. aureus*. We also present data showing that PUFAs and aminoglycosides effectively kill *S. aureus* small colony variants, strains that are particularly recalcitrant to killing by many antibiotics. We conclude that cotreatment with PUFAs, which are molecules with low host toxicity, and aminoglycosides decreases the aminoglycoside concentration necessary to kill *S. aureus*, lowering the toxic side effects to the host associated with prolonged aminoglycoside exposure.

**IMPORTANCE** *Staphylococcus aureus* infects every niche of the human host, and these infections are the leading cause of Gram-positive sepsis. Aminoglycoside antibiotics are inexpensive, stable, and effective against many bacterial infections. However, *S. aureus* can shift its metabolism to become tolerant of aminoglycosides, requiring increased concentrations and/or longer courses of treatment, which can cause severe host toxicity. Here, we report that polyunsaturated fatty acids (PUFAs), which have low host toxicity, disrupt the *S. aureus* membrane, making the pathogen susceptible to aminoglycosides. Additionally, cotreatment with aminoglycosides is effective at killing *S. aureus* small colony variants, strains that are difficult to treat with antibiotics. Taken together, the data presented herein show the promise of PUFA cotreatment to increase the efficacy of aminoglycosides against *S. aureus* infections and decrease the risk to the human host of antibiotic-induced toxicity.

**KEYWORDS** MRSA, PUFA, *Staphylococcus aureus*, aminoglycoside, antibiotic tolerance, arachidonic acid, gentamicin, linoleic acid, persister, polyunsaturated fatty acid, small colony variant

Address correspondence to Eric P. Skaar, eric.skaar@vumc.org.

The authors declare no conflict of interest.

*S*taphylococcus aureus can infect every niche of the human host, causing infections ranging from minor skin lesions to sepsis and death (1). Aminoglycoside antibiotics rely on the proton motive force (PMF) to enter the cell (2). As a facultative anaerobe, *S. aureus*

obviates the need for PMF by generating energy through fermentation, leading to persisters that are tolerant of but not resistant to aminoglycosides (3–6). Fermenting *S. aureus* requires higher concentrations of aminoglycosides to eradicate infection, but these elevated concentrations lead to host nephrotoxicity and ototoxicity (7–9). A recent report showed that disruption of the *S. aureus* membrane prevents aminoglycoside tolerance (6, 10). Here, we report that polyunsaturated fatty acids (PUFAs), major components of the host inflammatory burst (11), synergize with aminoglycosides to kill *S. aureus*.

PUFAs contain multiple *cis*-double bonds, which increase membrane fluidity in *S. aureus* when incorporated into the phospholipid membrane (12–15). We therefore tested the hypothesis that cotreatment with PUFAs and the aminoglycoside gentamicin would increase antimicrobial efficacy. Cotreatment of *S. aureus* JE2 (16) with arachidonic acid (AA) (Fig. 1B, inset) and gentamicin resulted in no observable growth 24 h posttreatment (Fig. 1A). The bactericidal cotreatment of AA with gentamicin decreased viable *S. aureus* by 5 orders of magnitude compared to either individual treatment (Fig. 1B). Additionally, AA decreased the gentamicin MIC from $37 \pm 22$ $\mu$g/mL to $1.7 \pm 0.5$ $\mu$g/mL (Fig. S1E in the supplemental material). Palmitic acid (PA) is fully saturated (Fig. 1D, inset) and does not increase membrane fluidity, and no synergy was observed when palmitic acid was used as a cotreatment with gentamicin (Fig. 1C and D). Linoleic acid (LA), another abundant host PUFA, also synergized with gentamicin to kill *S. aureus* (Fig. 1E and F). The synergy between LA and gentamicin was less than that observed with AA and gentamicin. This result likely arose from the presence of two fewer *cis*-double bonds on LA (Fig. 1F, inset) and, thus, decreased membrane fluidity compared to the membrane fluidity conferred by AA. Collectively, these data demonstrate that PUFAs synergize with gentamicin to kill *S. aureus*.

These findings are not specific to gentamicin, as tobramycin, another aminoglycoside, also synergized with AA to kill *S. aureus* (Fig. S1A). No synergy was observed when the non-aminoglycoside protein synthesis inhibitors chloramphenicol (Fig. S1B) and erythromycin (Fig. S1C) were used as cotreatments with AA. JE2 is a methicillin-resistant *S. aureus* (MRSA) strain (16), so we tested whether cotreatment with AA and a $\beta$-lactam antibiotic, oxacillin, would alter its resistance, which it did not (Fig. S1D). These findings demonstrate that the observed synergy is not a general phenotype between antibiotics and PUFAs but is specific for aminoglycosides.

We also evaluated the effects of PUFA cotreatment on other *S. aureus* strains. The MRSA strains UAMS-1 (Fig. S2A) (17), USA300 (Fig. S2B) (18), and MW2 (Fig. S2C) (19) were equally inhibited by the concentration of AA used. Strain Newman (Fig. S2D) (20), which is methicillin susceptible, was much more susceptible to AA than other *S. aureus* strains tested, consistent with previous studies (14, 15). However, methicillin resistance did not determine gentamicin tolerance, as all the strains tested were equally tolerant of the concentration of gentamicin used. Regardless of methicillin susceptibility, AA and gentamicin synergized to kill all the *S. aureus* strains tested, indicating that AA cotreatment may be a viable therapeutic strategy for infections caused by most *S. aureus* strains.

*S. aureus* persister cell formation is characterized by a shift in metabolism from respiration to fermentation, collapse of the PMF, and tolerance of aminoglycosides (3–6). Consistent with this, infections caused by *S. aureus* strains that are respiration deficient due to genetic mutations or small colony variants (SCVs) are notoriously difficult to treat with antibiotics, including aminoglycosides (21–24). Therefore, we hypothesized that cotreatment with AA and gentamicin would restore gentamicin activity against *S. aureus* SCVs. *S. aureus* strain NE1345, from the Nebraska Transposon Mutant Library (16), has a transposon insertion in *menD* that inactivates an enzyme essential for menaquinone biosynthesis. Menaquinone is an electron transport chain molecule essential for respiration, and an inability to produce menaquinone forces the organism to use fermentation for growth (21). Inactivation of *menD* impairs the growth of *S. aureus*, which can be complemented with exogenous menadione (Fig. 2A), a precursor used to synthesize menaquinone (25). A higher gentamicin concentration was required to achieve growth inhibition of strain NE1345 (Fig. 2C) due to the impaired PMF. This concentration of gentamicin was also unable to completely kill *S. aureus* JE2 in the absence of AA (Fig. 2B). Cotreatment of NE1345 with AA and gentamicin resulted

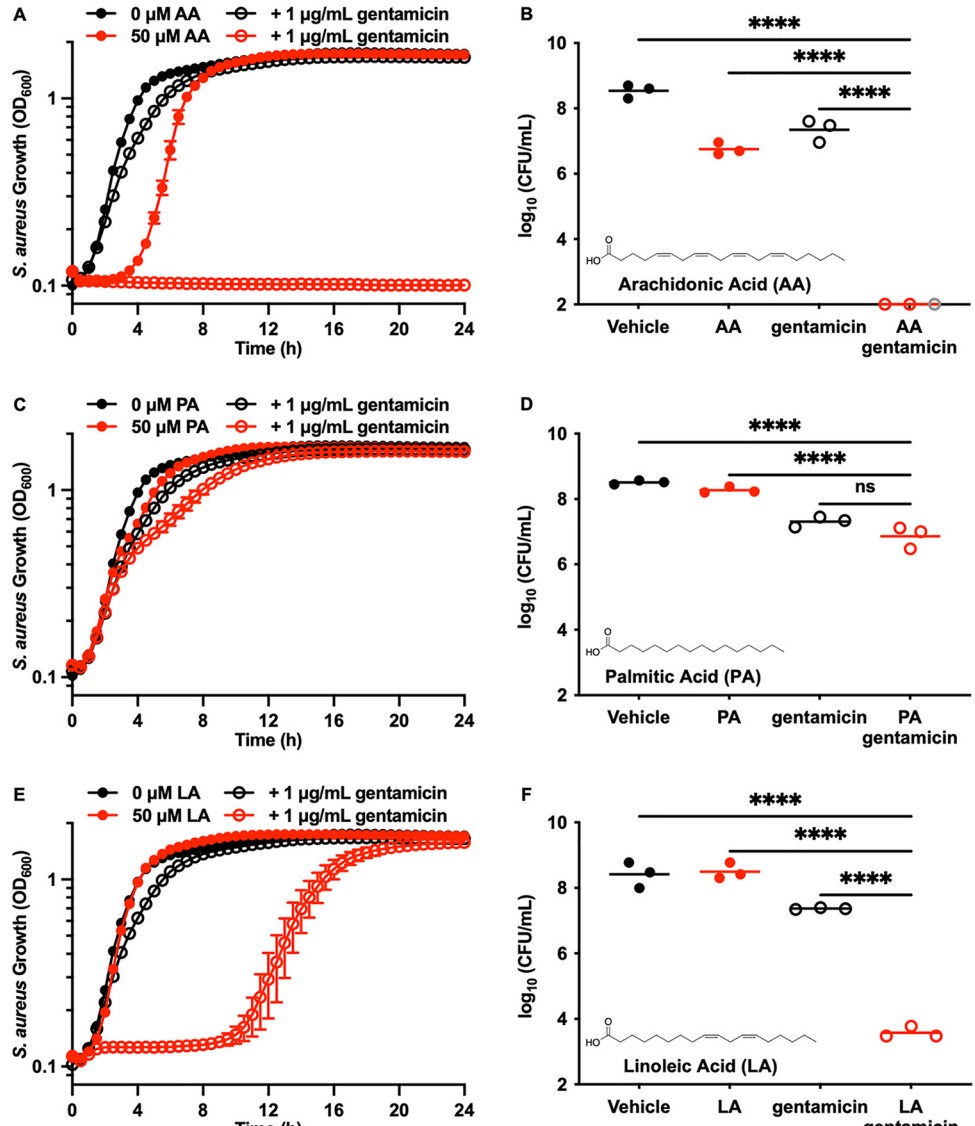

**FIG 1** Polyunsaturated fatty acids potentiate gentamicin activity against *S. aureus*. (A) JE2 was treated with vehicle, 1 μg/mL gentamicin, 50 μM AA, or gentamicin and AA. Bacterial growth was monitored by optical density at 600 nm (OD$_{600}$) every 30 min for 24 h. Data are the mean values ± standard errors of the means for measurements acquired in biological triplicate. (B) JE2 was treated with vehicle, 1 μg/mL gentamicin, 50 μM AA, or gentamicin and AA for 3 h at 37°C. After incubation, viable bacteria were quantified by dilution plating on solid medium. Data are presented as mean values, and each point represents a single biological replicate. Gray symbols represent replicates with the number of viable bacterial colonies below the limit of detection. (C) JE2 was treated with vehicle, 1 μg/mL gentamicin, 50 μM PA, or gentamicin and PA. Bacterial growth was monitored by OD$_{600}$ every 30 min for 24 h. Data are mean values ± standard errors of the means for measurements acquired in biological triplicate. (D) JE2 was treated with vehicle, 1 μg/mL gentamicin, 50 μM PA, or gentamicin and PA for 3 h at 37°C. After incubation, viable bacteria were quantified by dilution plating on solid medium. Data are presented as mean values, and each point represents a single biological replicate. (E) JE2 was treated with vehicle, 1 μg/mL gentamicin, 50 μM LA, or gentamicin and LA. Bacterial growth was monitored by OD$_{600}$ every 30 min for 24 h. Data are mean values ± standard errors of the means for measurements acquired in biological triplicate. (F) JE2 was treated with vehicle, 1 μg/mL gentamicin, 50 μM LA, or gentamicin and LA for 3 h at 37°C. After incubation, viable bacteria were quantified by dilution plating on solid medium. Data are presented as mean values, and each point represents the value for a single biological replicate. *P* values were calculated by one-way analysis of variance (ANOVA). Nonsignificant (ns), *P* > 0.05; ****, *P* < 0.0001.

in no observable growth following treatment (Fig. 2C). Treatment of NE1345 with gentamicin alone decreased viable bacteria by 7-fold, while cotreatment with AA and gentamicin decreased viable bacteria by over 25,000-fold compared to the results for vehicle-treated NE1345 (Fig. 2D). Cotreatment with AA and gentamicin worked synergistically to kill MRSA strain 5005 (Fig. S2E), a gentamicin-susceptible clinical isolate. However, no synergy was

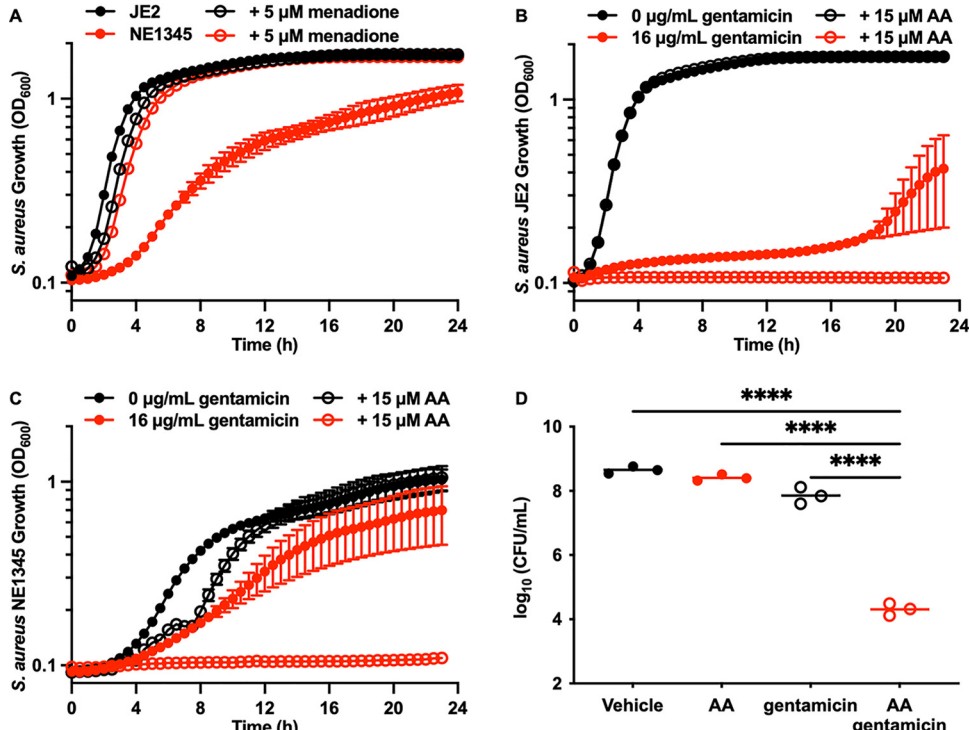

**FIG 2** Cotreatment with AA and gentamicin kills *S. aureus* small colony variants. (A) JE2 and NE1345 (*menD*) were treated with vehicle or 5 $\mu$M menadione. Bacterial growth was monitored by $OD_{600}$ every 30 min for 24 h. Data are mean values $\pm$ standard errors of the means for measurements acquired in biological triplicate. (B) JE2 was treated with vehicle, 15 $\mu$M AA, 16 $\mu$g/mL gentamicin, or gentamicin and AA. Bacterial growth was monitored by $OD_{600}$ every 30 min for 24 h. Data are mean values $\pm$ standard errors of the means for measurements acquired in biological triplicate. (C) NE1345 (*menD*) was treated with vehicle, 15 $\mu$M AA, 16 $\mu$g/ mL gentamicin, or gentamicin and AA. Bacterial growth was monitored by $OD_{600}$ every 30 min for 24 h. Data are mean values $\pm$ standard errors of the means for measurements acquired in biological triplicate. (D) NE1345 (*menD*) was treated with vehicle, 15 $\mu$M AA, 16 $\mu$g/mL gentamicin, or gentamicin and AA for 18 h at 37°C. After incubation, viable bacteria were quantified by dilution plating on solid medium. Data are presented as mean values, and each point represents the value for a single biological replicate. *P* values were calculated by one-way ANOVA. ****, $P < 0.0001$.

observed in cotreatment of MRSA strain 10554, a gentamicin-resistant clinical isolate. MRSA 10554 expresses the enzyme Aac(6′)-Aph(2″), which acetylates the 6′ amine and phosphorylates the 2″ hydroxyl of gentamicin, decreasing the ability of gentamicin to bind the ribosome (26). The lack of synergy between AA and gentamicin in MRSA 10554 was anticipated because resistance is obtained through modification of gentamicin, as opposed to the decreased gentamicin uptake seen with NE1345. These data demonstrate that, while the combination of PUFAs and aminoglycosides cannot overcome resistance mechanisms that inactivate aminoglycosides directly, they can restore antibiotic susceptibility in cases of tolerance, as seen with SCVs.

Worldwide, aminoglycosides are an important class of antibiotics, but host toxicity and the ability of *S. aureus* to tolerate high concentrations decreases their efficacy and prevents their widespread use for treatment of these infections. We report that PUFAs synergize with aminoglycosides to kill *S. aureus*, including SCVs, which are difficult to eradicate with antibiotics. These findings open the potential of using PUFAs to lower the concentration of aminoglycoside and/or shorten the treatment duration necessary to cure *S. aureus* infections. While the delivery of AA systemically will be a challenge due to uptake and metabolism by the host, topical treatments that combine AA and aminoglycosides for staphylococcal skin infections in humans (1) and mastitis in dairy livestock (27) can be tested in the near future. Furthermore, we add to the existing research showing that membrane-disrupting molecules with low host toxicity can be screened for synergy with aminoglycosides as a method to discover new cotreatments that eradicate *S. aureus* and lower the risk of harmful antibiotic side effects to patients.

## SUPPLEMENTAL MATERIAL

Supplemental material is available online only.

**SUPPLEMENTAL FILE 1**, PDF file, 0.4 MB.

## ACKNOWLEDGMENTS

We thank James J. Galligan and Andrew J. Monteith for critical reading and feedback on the manuscript. We also thank Rebecca C. Christofferson and Juan J. Martinez for generously supplying common laboratory equipment, helping us circumvent backordered supplies. *S. aureus* strains JE2 and NE1345 were provided by the Network on Antimicrobial Resistance in *Staphylococcus aureus* (NARSA) for distribution by BEI Resources, NIAID, NIH: Nebraska Transposon Mutant Library (NTML).

We also thank the sources of the funding that made the manuscript possible: grants number R01 AI069233 (E.P.S.), R01AI073843 (E.P.S.), R01AI150701 (E.P.S.), T32 ES007028 (M.J.M.), T32 AI007474 (J.A.F.), and 18POST34030426 (W.N.B.).

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
