## [Reviewer comments · Microbiology Spectrum]

Microbiology Spectrum

Host polyunsaturated fatty acids potentiate aminoglycoside killing of *Staphylococcus aureus*

William Beavers, Matthew Munneke, Alex Stackhouse, Jeffrey Freiberg, and Eric Skaar

Corresponding Author(s): Eric Skaar, Vanderbilt University Medical Center

Review Timeline:

Submission Date:	January 4, 2022
Editorial Decision:	January 31, 2022
Revision Received:	March 8, 2022
Accepted:	March 15, 2022

Editor: Amanda Oglesby

Reviewer(s): The reviewers have opted to remain anonymous.

Transaction Report:

DOI: <https://doi.org/10.1128/spectrum.02767-21>

January 31, 2022

Dr. Eric P Skaar
Vanderbilt University Medical Center
Pathology, Microbiology, and Immunology
1161 21st Avenue South
MCN A5211
Nashville, TN 37232

Re: Spectrum02767-21 (Host polyunsaturated fatty acids potentiate aminoglycoside killing of *Staphylococcus aureus*)

Dear Dr. Eric P Skaar:

Thank you for submitting your manuscript to Microbiology Spectrum. Your manuscript was reviewed by two experts, both of whom were very positive about this study. Reviewer 1 has made some suggestions for improvement that I would like you to consider. The suggested experiments are not critical for acceptance, but they would obviously strengthen the manuscript if they are feasible.

Link Not Available

Sincerely,

Amanda Oglesby

Journals Department
Reviewer comments:

Reviewer #1 (Comments for the Author):

This well written and informative manuscript by Beavers et al. documents the synergistic activity between arachidonic acid and aminoglycosides with *S. aureus*. It was documented using growth analysis that a combination of arachidonic acid and gentamicin is synergistic against *S. aureus*. Further, it was documented this combination was also very active against staphylococcal persisters. I just have a few comments that may improve the manuscript.

1. The authors should use the word susceptible throughout instead of sensitive.
2. The investigators may want to perform some standard MIC assays/synergy assays to document the synergistic activity of the two compounds. It is clear from the growth assays that they are synergistic, but these analyses would make the data more interpretable and relatable to other studies assessing synergy.

3. It would also seem relevant to assess if synergy occurs with an isolate that is resistant to aminoglycosides via expression of an aminoglycoside modifying enzyme. This would be a clinically relevant assessment.

4. Do the investigators believe that arachidonic acid can be utilized as a treatment paradigm with aminoglycosides? Or would they be rapidly metabolized by the human host?

Reviewer #2 (Comments for the Author):

The manuscript by Beavers et al. describes how PUFAs can potentiate aminoglycoside activity against *S. aureus*. This work builds on previous studies examining how membrane acting agents can induce aminoglycoside uptake independently of the proton-motive force. The ability of PUFAs to similarly potentiate aminoglycosides is important and may influence the development of novel therapeutic approaches. The paper is well written, the results are impressive and they have been appropriately interpreted.

Staff Comments:

Preparing Revision Guidelines

Please return the manuscript within 60 days; if you cannot complete the modification within this time period, please contact me. If you do not wish to modify the manuscript and prefer to submit it to another journal, please notify me of your decision immediately so that the manuscript may be formally withdrawn from consideration by Microbiology Spectrum.

We thank the Reviewers for their thoughtful and thorough review of this manuscript and think that the resulting manuscript is much improved. Below is a point-by-point, detailed description of all of the changes made to the manuscript in response to the following reviewer comments.

Reviewer #1 (Comments for the Author):

This well written and informative manuscript by Beavers et al. documents the synergistic activity between arachidonic acid and aminoglycosides with *S. aureus*. It was documented using growth analysis that a combination of arachidonic acid and gentamicin is synergistic against *S. aureus*. Further, it was documented this combination was also very active against staphylococcal persisters. I just have a few comments that may improve the manuscript.

1. The authors should use the word susceptible throughout instead of sensitive.

Susceptible is now used in place of sensitive in the manuscript.

2. The investigators may want to perform some standard MIC assays/synergy assays to document the synergistic activity of the two compounds. It is clear from the growth assays that they are synergistic, but these analyses would make the data more interpretable and relatable to other studies assessing synergy.

To address this comment, we determined the gentamicin MIC for *S. aureus* JE2 +/- 50 μ M AA. Co-treatment of AA with gentamicin decreases the MIC by greater than 20-fold (Figure S1E).

3. It would also seem relevant to assess if synergy occurs with an isolate that is resistant to aminoglycosides via expression of an aminoglycoside modifying enzyme. This would be a clinically relevant assessment.

We obtained two clinical isolates from a collaborator. MRSA 5005 is susceptible to gentamicin, and synergy is observed when AA and gentamicin are co-treated (Figure S2E). MRSA 10554 is resistant to gentamicin through the gene *aac(6')-aph(2'')*, which encodes for a bifunctional gentamicin modifying enzyme. Aac(6')-Aph(2'') modification of gentamicin by acetylation of the 6' amine or phosphorylation of the 2'' hydroxyl, prevents synergy between AA and gentamicin because this resistance mechanism alters the ability of gentamicin to bind the ribosome, while the synergy described in this manuscript alters the ability of gentamicin to enter the *S. aureus* cell (Figure S2F).

4. Do the investigators believe that arachidonic acid can be utilized as a treatment paradigm with aminoglycosides? Or would they be rapidly metabolized by the human host?

A sentence was added to the concluding paragraph acknowledging the difficulties in delivering AA systemically in the host due to uptake and metabolism, and emphasizing that using the combination treatment of AA and gentamicin for skin infections can be attempted in experimental models immediately.

Reviewer #2 (Comments for the Author):

The manuscript by Beavers et al. describes how PUFAs can potentiate aminoglycoside activity against *S. aureus*. This work builds on previous studies examining how membrane acting agents can induce aminoglycoside uptake independently of the proton-motive force. The ability of PUFAs to similarly potentiate aminoglycosides is important and may influence the development of novel therapeutic approaches. The paper is well written, the results are impressive and they have been appropriately interpreted.

March 15, 2022

Dr. Eric P Skaar
Vanderbilt University Medical Center
Pathology, Microbiology, and Immunology
1161 21st Avenue South
MCN A5211
Nashville, TN 37232

Re: Spectrum02767-21R1 (Host polyunsaturated fatty acids potentiate aminoglycoside killing of *Staphylococcus aureus*)

Dear Dr. Eric P Skaar:

Your manuscript has been accepted, and I am forwarding it to the ASM Journals Department for publication. You will be notified when your proofs are ready to be viewed.

Sincerely,

Amanda Oglesby
Editor, Microbiology Spectrum
